# Protein conformational transitions explored by a morphing approach based on normal mode analysis in internal coordinates

**Byung Ho Lee**[1], **Soon Woo Park**[1], **Soojin Jo**[2], **Moon Ki Kim**[1,3]*

1 School of Mechanical Engineering, Sungkyunkwan University, Suwon, South Korea, 2 Department of Physics and Institute of Basic Science, Sungkyunkwan University, Suwon, South Korea, 3 Sungkyunkwan Advanced Institute of Nanotechnology (SAINT), Sungkyunkwan University, Suwon, South Korea

* mkkim1212@skku.edu

## Abstract

Large-scale conformational changes are essential for proteins to function properly. Given that these transition events rarely occur, however, it is challenging to comprehend their underlying mechanisms through experimental and theoretical approaches. In this study, we propose a new computational methodology called internal coordinate normal mode-guided elastic network interpolation (ICONGENI) to predict conformational transition pathways in proteins. Its basic approach is to sample intermediate conformations by interpolating the interatomic distance between two end-point conformations with the degrees of freedom constrained by the low-frequency dynamics afforded by normal mode analysis in internal coordinates. For validation of ICONGENI, it is applied to proteins that undergo open-closed transitions, and the simulation results (i.e., simulated transition pathways) are compared with those of another technique, to demonstrate that ICONGENI can explore highly reliable pathways in terms of thermal and chemical stability. Furthermore, we generate an ensemble of transition pathways through ICONGENI and investigate the possibility of using this method to reveal the transition mechanisms even when there are unknown metastable states on rough energy landscapes.

## 1 Introduction

Max Perutz and John Kendrew first determined the three-dimensional (3D) structures of hemoglobin and myoglobin in the 1960s, which laid the foundation for the field of structural biology [1–3]. Since then, numerous experiment-based studies have been performed to reveal structural information of macromolecules, resulting in more than 183,000 atomic-level structures in the Protein Data Bank (PDB) archive [4]. In addition, the vast array of information has demonstrated that regulated conformational changes are of crucial importance for proteins to perform their biological functions, which has led to increasing awareness of the need to probe these large transitions. Indeed, various experimental techniques such as nuclear magnetic resonance spectroscopy [5], small-angle X-ray scattering [6], and single-molecule spectroscopy [7] have been widely utilized to analyze the dynamic behavior of proteins. However,

**Data Availability Statement:** All relevant data are available at http://bioengineering.skku.ac.kr/kosmos/download.php.

**Funding:** BHL, SWP, and MKK are supported by Basic Science Research Program through the

National Research Foundation of Korea (NRF) funded by the Ministry of Science and ICT (No. 2019R1F1A1042633) and by the Ministry of Education (No. 2021R1A6A1A03039696). SJ is supported by Basic Science Research Program through the National Research Foundation of Korea (NRF) funded by the Ministry of Education (No. 2020R1I1A1A01055405). The funders had no role in study design, data collection and analysis, decision to publish, or preparation of the manuscript.

**Competing interests:** The authors have declared that no competing interests exist.

obtaining experimental information on the conformational changes of proteins is a longstanding challenge due to not only the intrinsic properties of the transition events with short-lived intermediate conformations, but also several technical limitations like sample preparation, system size, and time scale [8, 9].

Aside from the experimental studies, computational methods have played a key role in better understanding the functionally relevant dynamics of proteins that are difficult to capture through experimental approaches. Especially, molecular dynamics (MD) simulation, which samples conformational states in atomic detail by calculating interatomic forces using molecular mechanics force fields, has become one of the most powerful and popular tools [10, 11]. However, despite its successful applications in numerous studies, MD simulation has intrinsic limitations in exploring the large-scale conformational changes: the simulated systems easily get trapped in stable or metastable states and rarely cross high-energy barriers toward functional states, even on millisecond time scales. Recently, various MD strategies, such as development of special-purpose supercomputers [12, 13] and enhanced sampling methods [14–16], have contributed greatly to improving the performance of MD simulation, but the time-scale limitation remains to be resolved.

As an alternative approach to overcome the issue of computational complexity, normal mode analysis (NMA) has received much attention because it provides an efficient way to elucidate the intrinsic dynamics of proteins that are related to the global transitions [17–20]. NMA calculation is based on harmonic approximation of the potential energy function, and the resulting mode shapes are valid only near an equilibrium state. In other words, this method has inherent limitations in directly predicting conformational transitions that require inharmonic movements over energy barriers. Therefore, various methods combining NMA with other computational techniques have been developed to explore effective transition pathways between two end-point conformations [21–24].

In this study, we propose a new NMA-based pathway generation method called internal coordinate normal mode-guided elastic network interpolation (ICONGENI), an improved technique over the normal mode-guided elastic network interpolation (NGENI) [25]. The fundamental concept of both methods is to obtain intermediate conformations comprising a transition pathway by iteratively calculating displacement vectors to minimize error between the simulated intermediates and the targeted ones. In this process, NMA calculation is required to represent the displacement vectors as linear combinations of the lowest normal mode shapes and makes a critical difference between the two techniques: it is achieved in internal coordinates (IC-NMA) and Cartesian coordinates (CC-NMA) in ICONGENI and NGENI, respectively. CC-NMA has been widely used in studying protein dynamics due to the inherent nature of Cartesian coordinates (CCs): high computational efficiency and intuitive expression of protein dynamics but has the disadvantage of producing the mode shapes having unrealistic distortions like bond length stretching and bond angle bending [26]. On the other hand, IC-NMA has a distinctive advantage in describing the large-scale transitions in proteins. Internal coordinates (ICs) inherently facilitate the separation of the torsion angles from the others, so that IC-NMA can be performed in torsion angle space. Given that the conformational changes are dominantly influenced by the variations in the torsion angles, not in the bond lengths and the bond angles that are nearly rigid, this strategy enables it to produce chemically relevant mode shapes preventing the unrealistic distortions of bond lengths and bond angles and extending the validity of the harmonic approximation in calculation [26–28]. In terms of computational complexity, IC-NMA is less efficient than CC-NMA because it has extra calculations involved in transformation either from CCs to ICs or from ICs to CCs (see section 2.2 for further details), but this is not a critical issue because both methods can be performed at the personal computer level. In other words, IC-NMA is more suitable than CC-NMA for

describing and exploring the conformational changes in proteins, which provides an insight into the development of ICONGENI.

For validation of ICONGENI, we first demonstrate the superiority of IC-NMA in predicting large-scale conformational transitions by comparing the performance of ICONGENI to that of NGENI where the pathway is explored using CC-NMA [25]. Both methods are applied to two proteins: *E. coli* adenylate kinase (ADK) and *E. coli* ribose-binding protein (RBP). The comparative analyses of the distributions of ICs and the potential energies of the resulting pathways show that the transition pathways simulated by ICONGENI have higher thermal and chemical stability than those by NGENI. However, its efficient manner of computing intermediate structures has intrinsic limitations in exploring large-scale transitions on complex energy landscapes. To address this issue, ICONGENI generated a pathway ensemble for ADK dependent on the number of normal modes used in these simulations and characterized the ensemble in interdomain angle space, demonstrating that ICONGENI can explore plausible pathways on complex free energy landscapes.

## 2 Materials and methods

### 2.1 Protein structural information

To explore transition pathways of proteins through ICONGENI and NGENI, two end-point structures of each protein should be used as reference information. In the ADK case, the open and closed structures are chain A in PDB entry 4AKE (4AKE:A) [29] and 1AKE:A [30], respectively. In the RBP case, the open and closed structures are 1BA2:A [31] and 2DRI:A [32], respectively. In addition, we used several experimental intermediate structures of ADK whose PDB code 1ZIN, 1ZIO, 1ZIP [33], and 1DVR [34] to experimentally evaluate the ICONGENI simulation results. Because these intermediate structures have similar conformations, but different sequences with the reference structures (*i.e.*, 4AKE:A and 1AKE:A), homology modeling was implemented using Modeller v9.25 [35]. In detail, 10 candidate models of the intermediate structures were constructed by using their 3D conformations (as templates) and the 2D sequences of the reference structures (as target proteins), and the best models for each template were selected based on DOPE score [36]. The selected structures were refined by energy minimization (500 steps of conjugate gradient) using the CHARMM36m force field [37].

### 2.2 Internal coordinate normal mode-guided elastic network interpolation (ICONGENI)

The ultimate goal of ICONGENI is to predict pathways for large-scale conformational changes of macromolecules based on structural information on two end-point (*i.e.*, initial and final) conformations. To do this, valid displacement vectors can be obtained iteratively through ICONGENI, leading to determination of the consecutive intermediate conformations that comprise the pathways. A brief explanation of its algorithm is as follows.

First, IC-NMA is required prior to any procedure because we assume that the displacement vectors are linear combinations of low-frequency normal mode vectors. Next, the cost function is constructed to compute the degree of difference in interatomic distance between the resulting and desired conformations. Using the cost function, we devise compromise solutions (*i.e.*, a series of weights assigned to the normal modes of the displacement vectors) between minimizing the values of the cost function and constraining the degrees of freedom (DOFs) of the structural dynamics with some low-frequency mode shapes. By repeating these cycles, promising transition pathways can be predicted. A detailed explanation of ICONGENI will be introduced in the following subsections.

**2.2.1 Elastic network model (ENM).** ICs characterize molecular geometry using bond lengths, bond angles, and torsion angles terms to facilitate a better understanding of the

structural dynamics of molecules. While the backbone bond lengths and bond angles are nearly fixed in molecular systems, some torsion angles (phi ($\phi$) and psi ($\psi$) angles in the protein conformation) can vary, and their dynamics exert a strong influence on the large-scale conformational changes. To effectively describe molecular systems in ICs, we use a coarse-grained modeling method, the elastic network model (ENM), wherein specific atoms of a protein backbone (*i.e.*, N, $C_\alpha$, and C) are sampled and linked by a unit spring constant. The spring constant $k$ matrix is defined as

$$k_{i,j} = \begin{cases} 0; & d > d_{cutoff} \\ 1; & d \leq d_{cutoff} \end{cases} \tag{1}$$

where $k_{i,j}$ is a binary spring constant between atoms $i$ and $j$, $d$ is an actual distance between them, and $d_{cutoff}$ is a cutoff distance set to be 12 Å [19].

**2.2.2 Normal mode analysis in internal coordinates (IC-NMA).** For IC-NMA calculation, the second derivatives of the kinetic and potential energy functions are required. The main strategy is to calculate the functions with respect to CCs and then to transform them into those with respect to ICs. All calculations of IC-NMA will be described in this section with respect to previous works [38–41].

*2.2.2.1 Transformation from Cartesian to internal coordinates.* To obtain the second derivatives of the kinetic and potential energy functions, the first derivatives of CCs with respect to ICs must be defined. Only torsion angles are regarded as variables for the calculation, while the bond lengths and bond angles are considered to be fixed. The following derivation of the first derivatives will be based on some assumptions. First, the Eckart condition is assumed to separate the internal and external motions because the ICs cannot express external motions [42]. For the position vector $r_i$ of atom $i$, the condition can be satisfied by the following equations:

$$\sum_i m_i dr_i = 0 \tag{2}$$

$$\sum_i m_i r_i^{ref} \times dr_i = 0 \tag{3}$$

where $m_i$ and $r_i^{ref}$ are the mass and fixed position vector of atom $i$, respectively, in the reference conformation.

Next, let $\theta_\alpha$ be a torsion angle around chemical bond $\alpha$ and domains $A$ and $B$ be the two domains divided by bond $\alpha$ as shown in **Fig 1**. Then, the two domains are regarded as rigid bodies, based on which Eq (2) can be rewritten as

$$M_A dr_A + M_B dr_B = 0 \tag{4}$$

where $M_A = \sum_i^A m_i$, $M_B = \sum_i^B m_i$, and $r_A$ and $r_B$ are the position vectors of the center of mass of domains $A$ and $B$, respectively.

If $\omega_A$ and $\omega_B$ are the rotation vectors of domains $A$ and $B$, respectively, their relative rotation vectors $\omega_{AB}$ and $\delta r_i$ are defined as follows:

$$\omega_{AB} = d\theta_\alpha e_\alpha = \omega_B - \omega_A \tag{5}$$

$$dr_i = \begin{cases} dr_A + \omega_A \times (r_i^{ref} - r_A^{ref}); & i \in A \\ dr_B + \omega_B \times (r_i^{ref} - r_B^{ref}); & i \in B \end{cases} \tag{6}$$

where $e_\alpha = \frac{r_{t(\alpha)} - r_{t(\alpha)-1}}{\|r_{t(\alpha)} - r_{t(\alpha)-1}\|}$.

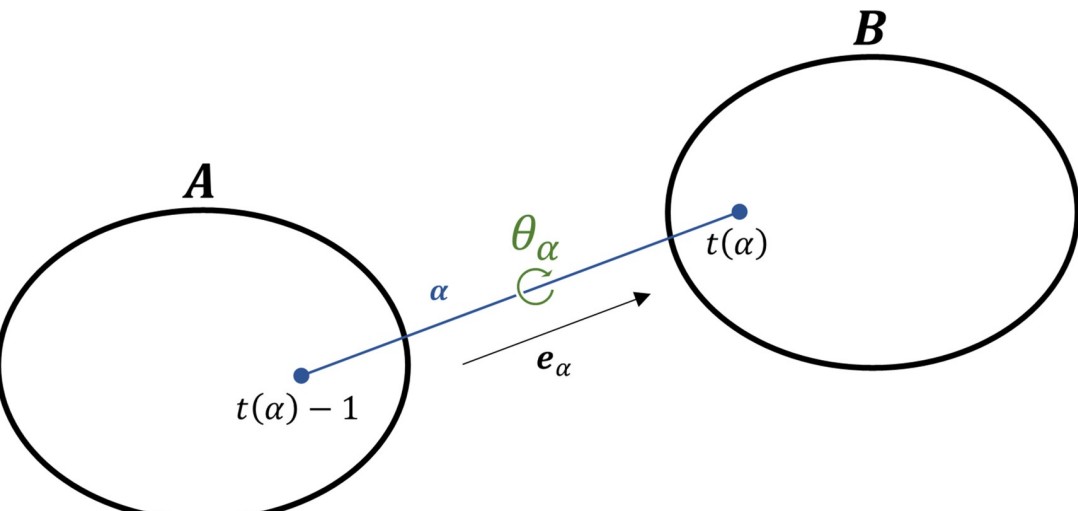

**Fig 1.** Schematic of a molecular system composed of two rigid bodies $A$ and $B$ with a chemical bond $\alpha$. The relative displacement between the two bodies can be defined by the torsion angle $\theta_\alpha$ around the bond $\alpha$. If a bond $\alpha$ links atoms $(i-1)$ to $i$, $t(\alpha)$ designates $i$.

Using Eq (5), $d\boldsymbol{r}_i$ in domain $B$ (corresponding to the second equation of Eq (6)) can be rewritten as

$$d\boldsymbol{r}_i = d\boldsymbol{r}_A + \boldsymbol{\omega}_A \times (\boldsymbol{r}_i^{ref} - \boldsymbol{r}_A^{ref}) + d\theta_\alpha \boldsymbol{e}_\alpha \times (\boldsymbol{r}_i^{ref} - \boldsymbol{r}_{t(\alpha)}^{ref}); \quad i \in B \tag{7}$$

From Eqs (6) and (7),

$$d\boldsymbol{r}_A + \boldsymbol{\omega}_A \times (\boldsymbol{r}_{t(\alpha)}^{ref} - \boldsymbol{r}_A^{ref}) = d\boldsymbol{r}_B + \boldsymbol{\omega}_B \times (\boldsymbol{r}_{t(\alpha)}^{ref} - \boldsymbol{r}_B^{ref}) \tag{8}$$

Subsequently, using Eqs (3) and (6) and the concept of angular momentum,

$$M_A \boldsymbol{r}_A \times (d\boldsymbol{r}_A - \boldsymbol{\omega}_A \times \boldsymbol{r}_A^{ref}) + M_B \boldsymbol{r}_B \times (d\boldsymbol{r}_B - \boldsymbol{\omega}_B \times \boldsymbol{r}_B^{ref}) + I_A \boldsymbol{\omega}_A + I_B \boldsymbol{\omega}_B = 0 \tag{9}$$

where the inertia tensors are given by $\boldsymbol{I}_A = \sum_i^A m_i \|\boldsymbol{r}_i^{ref}\|^2 \boldsymbol{E}_3 - \boldsymbol{r}_i^{ref}(\boldsymbol{r}_i^{ref})^T$ and $\boldsymbol{I}_B = \sum_i^B m_i \|\boldsymbol{r}_i^{ref}\|^2 \boldsymbol{E}_3 - \boldsymbol{r}_i^{ref}(\boldsymbol{r}_i^{ref})^T$. $\boldsymbol{E}_3$ is the 3×3 identity matrix.

From Eqs (4), (5), and (8),

$$d\boldsymbol{r}_A - \boldsymbol{\omega}_A \times \boldsymbol{r}_A^{ref} = d\theta_\alpha \boldsymbol{e}_\alpha \times \left\{ \left(1 - \frac{M_A}{M}\right)\boldsymbol{r}_{t(\alpha)}^{ref} + \frac{M_A}{M}\boldsymbol{r}_A^{ref} \right\} \tag{10}$$

where $M = M_A + M_B$.

Using Eqs (9) and (10), $\boldsymbol{\omega}_A$ is expressed as

$$\boldsymbol{\omega}_A = -d\theta_\alpha [\boldsymbol{I}^{-1}\{M_A \boldsymbol{r}_A^{ref} \times (\boldsymbol{e}_\alpha \times \boldsymbol{r}_{t(\alpha)}^{ref}) + (\boldsymbol{I} - \boldsymbol{I}_A)\boldsymbol{e}_\alpha\}] \tag{11}$$

where $\boldsymbol{I} = \boldsymbol{I}_A + \boldsymbol{I}_B$.

By substituting Eqs (10) and (11) into Eq (6), the derivative of CCs with respect to ICs is

$$
\frac{\partial \boldsymbol{r}_i}{\partial \theta_\alpha} = 
\begin{cases}
\boldsymbol{e}_\alpha \times \left\{ \left(1 - \dfrac{M_A}{M}\right) \boldsymbol{r}_{t(\alpha)}^{ref} + \dfrac{M_A}{M} \boldsymbol{r}_A^{ref} \right\} - \boldsymbol{I}^{-1} \left\{ M_A \boldsymbol{r}_A^{ref} \times (\boldsymbol{e}_\alpha \times \boldsymbol{r}_{t(\alpha)}^{ref}) + (\boldsymbol{I} - \boldsymbol{I}_A)\boldsymbol{e}_\alpha \right\} \times \boldsymbol{r}_i^{ref}; \quad i \in A \\
-\boldsymbol{e}_\alpha \times \left\{ \left(1 - \dfrac{M_B}{M}\right) \boldsymbol{r}_{t(\alpha)}^{ref} + \dfrac{M_B}{M} \boldsymbol{r}_B^{ref} \right\} + \boldsymbol{I}^{-1} \left\{ M_B \boldsymbol{r}_B^{ref} \times (\boldsymbol{e}_\alpha \times \boldsymbol{r}_{t(\alpha)}^{ref}) + (\boldsymbol{I} - \boldsymbol{I}_B)\boldsymbol{e}_\alpha \right\} \times \boldsymbol{r}_i^{ref}; \quad i \in B
\end{cases}
\tag{12}
$$

Finally, these equations can be rewritten in the form of matrix-vector multiplication:

$$
\frac{\partial \boldsymbol{r}_i}{\partial \theta_\alpha} =
\begin{cases}
(\boldsymbol{P}_i \quad \boldsymbol{E}_3)
\begin{bmatrix}
\boldsymbol{I}^{-1}\boldsymbol{I}_B & \boldsymbol{I}^{-1}(\boldsymbol{P}_B)^T \\
\dfrac{\boldsymbol{P}_B}{M} & \dfrac{M_B}{M}\boldsymbol{E}_3
\end{bmatrix}
\boldsymbol{d}_\alpha; \qquad i \in A \\[2.5em]
-(\boldsymbol{P}_i \quad \boldsymbol{E}_3)
\begin{bmatrix}
\boldsymbol{I}^{-1}\boldsymbol{I}_A & \boldsymbol{I}^{-1}(\boldsymbol{P}_A)^T \\
\dfrac{\boldsymbol{P}_A}{M} & \dfrac{M_A}{M}\boldsymbol{E}_3
\end{bmatrix}
\boldsymbol{d}_\alpha; \qquad i \in B
\end{cases}
\tag{13}
$$

where $\boldsymbol{P}_i = \boldsymbol{r}_i \times = \begin{bmatrix} 0 & -z_i & y_i \\ z_i & 0 & -x_i \\ -y_i & x_i & 0 \end{bmatrix}$, $\boldsymbol{P}_A = \sum_i^A m_i \boldsymbol{P}_i$, $\boldsymbol{P}_B = -\boldsymbol{P}_A$, and $\boldsymbol{d}_\alpha = [\boldsymbol{e}_\alpha, \boldsymbol{e}_\alpha \times \boldsymbol{r}_{t(\alpha)}]^T$.

*2.2.2.2 Construction of the second derivative of kinetic energy.* As shown in **Fig 2**, the kinetic energy $\boldsymbol{T}$ of the molecular system in ICs can be calculated using two torsion angle variables $\theta_\alpha$ and $\theta_\beta$ as follows:

$$
\boldsymbol{T} = \frac{1}{2}\sum_i m_i \dot{\boldsymbol{r}}_i^2 = \frac{1}{2}\sum_{\alpha,\beta} \underbrace{\left( \sum_i m_i \frac{\partial \boldsymbol{r}_i}{\partial \theta_\alpha} \cdot \frac{\partial \boldsymbol{r}_i}{\partial \theta_\beta} \right)}_{\boldsymbol{M}_{\alpha,\beta}} \dot{\theta}_\alpha \dot{\theta}_\beta
\tag{14}
$$

where $\boldsymbol{M}_{\alpha,\beta}$ is the second derivative of kinetic energy for bonds $\alpha$ and $\beta$.

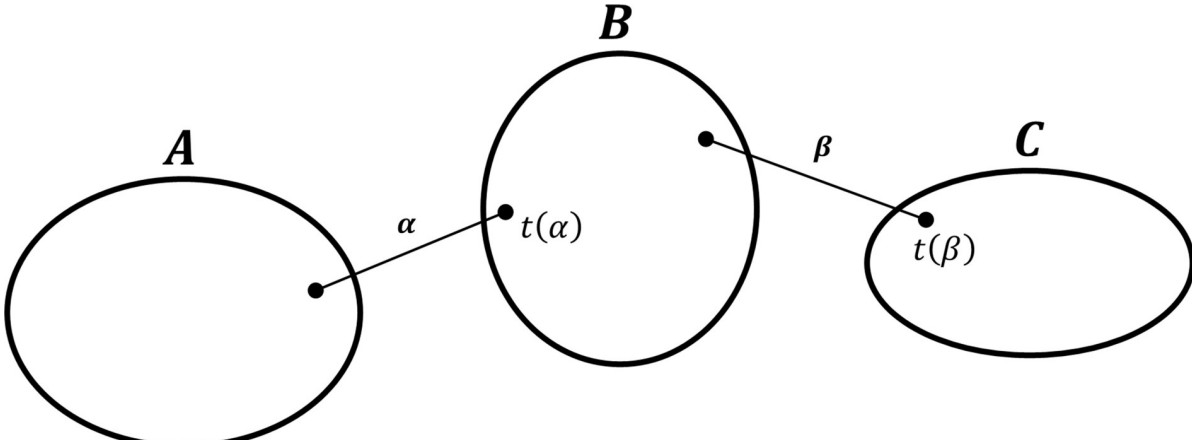

**Fig 2. Schematic of a molecular system composed of three rigid bodies $A$, $B$, and $C$ with two chemical bonds $\alpha$ and $\beta$.** If the bond $\alpha$ links atoms $(i-1)$ to $i$, $t(\alpha)$ designates $i$.

Then, using Eq (13), $M_{\alpha,\beta}$ can be defined as

$$
\begin{aligned}
M_{\alpha,\beta} &= (M_{\alpha,\beta})_A + (M_{\alpha,\beta})_B + (M_{\alpha,\beta})_C \\
&= (d_\alpha)^T \begin{pmatrix} I_A I^{-1} I_C + \dfrac{(P_A)^T P_C}{M} & I_A I^{-1} (P_C)^T + \dfrac{M_C}{M}(P_A)^T \\[2mm] P_A I^{-1} I_C + \dfrac{M_A}{M} P_C & P_A I^{-1} (P_C)^T + \dfrac{M_A M_C}{M} E_3 \end{pmatrix} d_\beta
\end{aligned}
\tag{15}
$$

where

$$
\begin{aligned}
(M_{\alpha,\beta})_A &= \sum_i^A m_i \frac{\partial r_i}{\partial \theta_\alpha} \cdot \frac{\partial r_i}{\partial \theta_\beta} \\
&= (d_\alpha)^T \begin{pmatrix} (I - I_A)I^{-1} & \dfrac{-(P_A)^T}{M} \\[2mm] -P_A I^{-1} & \dfrac{M - M_A}{M} E_3 \end{pmatrix} \begin{pmatrix} I_A & (P_A)^T \\[2mm] P_A & M_A E_3 \end{pmatrix} \begin{pmatrix} I^{-1} I_C & I^{-1}(P_C)^T \\[2mm] \dfrac{P_C}{M} & \dfrac{M_C}{M} E_3 \end{pmatrix} d_\beta,
\end{aligned}
$$

$$
(M_{\alpha,\beta})_B = -(d_\alpha)^T \begin{pmatrix} I_A I^{-1} & \dfrac{(P_A)^T}{M} \\[2mm] P_A I^{-1} & \dfrac{M_A}{M} E_3 \end{pmatrix} \begin{pmatrix} I_B & (P_B)^T \\[2mm] P_B & M_B E_3 \end{pmatrix} \begin{pmatrix} I^{-1} I_C & I^{-1}(P_C)^T \\[2mm] \dfrac{P_C}{M} & \dfrac{M_C}{M} E_3 \end{pmatrix} (d_\beta)^T,
$$

and

$$
(M_{\alpha,\beta})_C = (d_\alpha)^T \begin{pmatrix} I_A I^{-1} & \dfrac{(P_A)^T}{M} \\[2mm] P_A I^{-1} & \dfrac{M_A}{M} E_3 \end{pmatrix} \begin{pmatrix} I_C & (P_C)^T \\[2mm] P_C & M_C E_3 \end{pmatrix} \begin{pmatrix} I^{-1}(I - I_C) & -I^{-1}(P_C)^T \\[2mm] -\dfrac{P_C}{M} & \dfrac{M - M_C}{M} E_3 \end{pmatrix}.
$$

*2.2.2.3 Construction of the second derivative of potential energy.* Basically, the potential energy of a molecular system is defined based on interatomic interactions (*e.g.*, van der Waals bond and covalent bond potentials). With the aid of a simple assumption that one rigid-body domain is fixed instead of the Eckart condition, we can simply reformulate $\frac{\partial r_i}{\partial \theta_\alpha}$ against Eq (13). If domain *A* is fixed in **Fig 1**, the following equations are satisfied:

$$
\frac{\partial r_i}{\partial \theta_\alpha} = \begin{cases} 0; & i \in A \\ -(P_i \ E_3) d_\alpha; & i \in B \end{cases}
\tag{16}
$$

The second derivatives of potential energy in ICs can be derived from those in CCs as

$$
H_{\alpha,\beta} = \frac{\partial^2 V}{\partial \theta_\alpha \partial \theta_\beta} = \sum_{i,j} \frac{\partial r_i}{\partial \theta_\alpha} \cdot \underbrace{\frac{\partial^2 V}{\partial r_i \partial r_j}}_{H^c_{i,j}} \cdot \frac{\partial r_j}{\partial \theta_\beta}
\tag{17}
$$

where $V$ is the potential energy function, and $H_{\alpha,\beta}$ ($H^c_{i,j}$) is the second derivative of potential energy for the ICs $\theta_\alpha$ and $\theta_\beta$ (for the CCs $r_i$ and $r_j$).

$H_{i,j}^c$ is defined simply by the following function of interatomic distances [19]:

$$H_{i,j}^c = \frac{\partial^2 V}{\partial r_i \partial r_j} = -\frac{k_{i,j}}{\|r_i - r_j\|^2} \left(r_i - r_j\right)\left(r_i - r_j\right)^T \tag{18}$$

Then, supposing that domain $B$ is fixed in the diagram of **Fig 2** and using Eqs (16) and (18), $H_{\alpha,\beta}$ can be formulated as

$$H_{\alpha,\beta} = (d_\alpha)^T \left\{ \sum_i^A \sum_j^C \frac{k_{i,j}}{\|r_{i,j}\|^2} \begin{bmatrix} r_i \times r_j \\ r_i - r_j \end{bmatrix} \left[ (r_i \times r_j)^T \quad (r_i - r_j)^T \right] \right\} d_\beta \tag{19}$$

Using Eqs (15) and (19), we can construct the equation of motion with respect to ICs. For a molecular system having $n$ torsion angle dynamics,

$$M\ddot{\theta} + H\theta = 0 \tag{20}$$

where $\theta = [\theta_1 \quad \theta_2 \quad \cdots \quad \theta_n]$, $M = \begin{bmatrix} M_{1,1} & \cdots & M_{1,n} \\ \vdots & \ddots & \vdots \\ M_{n,1} & \cdots & M_{n,n} \end{bmatrix}$, and $H = \begin{bmatrix} H_{1,1} & \cdots & H_{1,n} \\ \vdots & \ddots & \vdots \\ H_{n,1} & \cdots & H_{n,n} \end{bmatrix}$.

We can obtain normal modes (*i.e.*, pairs of eigenvalues and eigenvectors) by solving a generalized eigenvalue problem for Eq (20). Next, an extra calculation is required to transform the resulting eigenvectors in ICs to those in CCs. Therefore, the final form of the normal mode vectors can be determined by the following equation [41]:

$$\Delta r_{i,k} = \sum_\alpha \frac{\partial r_i}{\partial \theta_\alpha} \Delta \theta_{\alpha,k} \tag{21}$$

where $\Delta r_{i,k}$ $(\Delta \theta_{\alpha,k})$ is the $k^{th}$ normal mode vector of atom $i$ (of bond $\alpha$) with respect to CCs (ICs).

**2.2.3 Construction of the cost function.** The cost function is defined as a function of error in interatomic distances between the simulated conformations and the desired ones [25, 43]:

$$C(\delta) = \frac{1}{2} \sum_{i,j} k_{i,j} \{\|(r_i + \delta_i) - (r_j + \delta_j)\| - q_{i,j}\}^2 \tag{22}$$

where $\delta_i$ is the displacement vector of atom $i$ describing conformational changes, and $q_{i,j}$ is the desired distance between atoms $i$ and $j$.

$q_{i,j}$ in a target intermediate can be determined through linear interpolation between the two end-point conformations as

$$q_{i,j} = (1-s)\|r_i^0 - r_j^0\| + s\|r_i^1 - r_j^1\| \tag{23}$$

where $r_i^0$ and $r_i^1$ represent the position vectors of atom $i$ for the initial and final conformations, respectively. $s$ is a proportional representation of the location of an intermediate to be simulated on the pathway when the total length of the pathway is set to 1.

In accordance with the strategy of ICONGENI, $\delta_i$ can be represented as the linear combination of a set of low-frequency normal mode vectors for reference conformations:

$$\delta_i = w_1 \Delta r_{i,1} + w_2 \Delta r_{i,2} + \cdots + w_m \Delta r_{i,m} \tag{24}$$

where $m$ denotes the number of low-frequency normal modes used in the simulation, and $w_n$ is the weight of the $n^{th}$ normal mode vector.

Using Eq (24), Eq (22) can be rewritten as

$$C(\boldsymbol{W}) = \frac{1}{2}\sum_{i,j}k_{i,j}\{\|(\boldsymbol{r}_i - \boldsymbol{r}_j) + (\boldsymbol{V}_i - \boldsymbol{V}_j)\boldsymbol{W}\| - q_{i,j}\}^2 \tag{25}$$

where $\boldsymbol{V}_i = [\Delta\boldsymbol{r}_{i,1} \ \Delta\boldsymbol{r}_{i,2} \ \cdots \ \Delta\boldsymbol{r}_{i,m}] \in R^{3\times m}$, and $\boldsymbol{W} = [w_1 \ w_2 \ \cdots \ w_m]^T \in R^{m\times 1}$.

Then, to find the value of $\boldsymbol{W}$ that minimizes $C(\boldsymbol{W})$, we simplify Eq (25) into the form of matrix-vector multiplication using a Taylor series expansion:

$$\|(\boldsymbol{r}_i - \boldsymbol{r}_j) + (\boldsymbol{V}_i - \boldsymbol{V}_j)\boldsymbol{W}\|$$
$$\approx \|\boldsymbol{r}_i - \boldsymbol{r}_j\| + \frac{(\boldsymbol{r}_i - \boldsymbol{r}_j)^T(\boldsymbol{V}_i - \boldsymbol{V}_j)\boldsymbol{W}}{\|\boldsymbol{r}_i - \boldsymbol{r}_j\|} + \frac{\boldsymbol{W}^T(\boldsymbol{V}_i - \boldsymbol{V}_j)^T\boldsymbol{A}(\boldsymbol{r}_i - \boldsymbol{r}_j)\{(\boldsymbol{V}_i - \boldsymbol{V}_j)\boldsymbol{W}\}}{2\|\boldsymbol{r}_i - \boldsymbol{r}_j\|} \tag{26}$$

where $\boldsymbol{A}\left(\boldsymbol{r}_i - \boldsymbol{r}_j\right) = \boldsymbol{E}_3 - \frac{(\boldsymbol{r}_i - \boldsymbol{r}_j)(\boldsymbol{r}_i - \boldsymbol{r}_j)^T}{\|\boldsymbol{r}_i - \boldsymbol{r}_j\|^2}$.

Then, Eq (27) can be written in the form of matrix-vector multiplication:

$$C(\boldsymbol{W}) = \boldsymbol{W}^T\boldsymbol{P}^{(1)}\boldsymbol{W} + \boldsymbol{P}^{(2)}\boldsymbol{W} + \boldsymbol{P}^{(3)} \tag{27}$$

where

$$\boldsymbol{P}^{(1)} = \frac{1}{2}\sum_{i=1}^{n-1}\sum_{j=i+1}^{n}k_{i,j}(\boldsymbol{V}_i - \boldsymbol{V}_j)^T\left\{\boldsymbol{E}_3 - q_{i,j}\frac{\boldsymbol{A}(\boldsymbol{r}_i - \boldsymbol{r}_j)}{\|\boldsymbol{r}_i - \boldsymbol{r}_j\|}\right\}(\boldsymbol{V}_i - \boldsymbol{V}_j),$$

$$\boldsymbol{P}^{(2)} = \frac{1}{2}\sum_{i=1}^{n-1}\sum_{j=i+1}^{n}k_{i,j}\left(2 - \frac{2q_{i,j}}{\|\boldsymbol{r}_i - \boldsymbol{r}_j\|}\right)(\boldsymbol{r}_i - \boldsymbol{r}_j)^T\left(\boldsymbol{V}_i - \boldsymbol{V}_j\right),$$

and

$$\boldsymbol{P}^{(3)} = \frac{1}{2}\sum_{i=1}^{n-1}\sum_{j=i+1}^{n}k_{i,j}\{(\boldsymbol{r}_i - \boldsymbol{r}_j)^T(\boldsymbol{r}_i - \boldsymbol{r}_j) - 2q_{i,j}\|\boldsymbol{r}_i - \boldsymbol{r}_j\| + q_{i,j}^2\}.$$

Finally, the optimal displacement vectors can be determined by solving for $\boldsymbol{W}$ from the following equation:

$$\frac{\partial C(\boldsymbol{W})}{\partial\boldsymbol{W}} = 2\boldsymbol{P}^{(1)}\boldsymbol{W} + \boldsymbol{P}^{(2)} = 0 \tag{28}$$

The optimal displacement vector allows us to determine the intermediate conformation on the pathway. By repeating this calculation process using the simulated intermediate as a new reference, the transition pathways from initial to final conformations can be generated. The number of iteration steps was determined so that consecutive intermediate structures differ by a root-mean-square deviation (RMSD) of about 0.1 Å. Here, we set the number of iteration steps for the case of ADK (RBP) is set to 71 (62) because the RMSD value between the two end-point conformations is 7.16 Å (6.25 Å).

## 3. Results and discussion

### 3.1 Comparing the ICONGENI pathways to NGENI pathways

In this section, we discuss the effectiveness of ICONGENI by comparing the resulting pathways to those developed by NGENI [25] under the same conditions. Because the main

difference between the two techniques is the coordinate space in which the NMA is performed (*i.e.*, ICONGENI and NGENI are based on IC-NMA and CC-NMA, respectively), it is expected that this comparative analysis will demonstrate the superiority of IC-NMA in describing large deformations of proteins. We performed ICONGENI and NGENI for two proteins: ADK and RBP. ADK as a phosphotransferase enzyme catalyzing the reaction ATP + AMP $\Leftrightarrow$ 2 ADP is composed of three domains: CORE, NMP, and LID, and undergoes two pairs of hinge motions of NMP and LID relative to CORE to fulfill its function [44]. RBP as one of the periplasmic binding proteins binds ribose through a hinge motion of two domains, which enables cells to sense and transport the ligand [31]. To predict the transition from open state to closed state, we obtained their 3D structures from PDB; the open and closed structures of ADK are chain A in PDB entry 4AKE:A and 1AKE:A, respectively, and those of RBP are 1BA2:A and 2DRI:A, respectively. The DOFs of these simulations were set to be the 50 lowest normal modes considered empirically sufficient to simulate the conformational changes within the experimental resolution based on our previous study [25]. For better understanding, the transition pathways explored by ICONGENI of ADK and RBP are provided in **S1** and **S2 Movies**, respectively.

First, the convergence issue of the pathways was addressed. To assess geometric convergence, we measured the RMSDs of the consecutive structures comprising the paths with respect to the final conformation (*i.e.*, the closed structures) and judged that the paths satisfied the convergence condition if the RMSD values steadily decreased below certain thresholds that is selected as the smaller of the experimental resolutions of two reference structures (*i.e.*, open and closed structures). As shown in **Fig 3A**, the ADK pathways of the two techniques had similar graphs and converged below the value of the resolution. Similarly, their RBP pathways also got close to the final conformation at a level beyond the resolution (**Fig 3B**). This confirmed that both techniques had no problem in terms of pathway convergence when generating the transition pathways based on the DOFs of the 50 lowest normal modes.

Next, we investigated backbone bond length and bond angle distributions on the simulated pathways. Proteins undergo conformational changes mainly through variations of two types of backbone torsion angles: $\phi$ around the $N{-}C_\alpha$ bond and $\psi$ around the $C_\alpha{-}C$ bond while variations of backbone bond lengths and bond angles are impractical during the transitions. The distributions of the bond lengths and bond angles provide key information to evaluate how well the proteins keep their molecular shape during large deformation. First, the backbone bond lengths are divided into three types: $N{-}C_\alpha$, $C_\alpha{-}C$, and $C{-}N$. According to the type, we calculated average (avg) and standard deviation (std) values over ICONGENI and NGENI pathways for two proteins (ADK and RBP) and analyzed their distributions using the experimental data as the reference avg and std values of the backbone bond length types (**Fig 4**) [45]. The avg values of the bond length in the ICONGENI paths were more concentrated around the corresponding experimental values than were those in the NGENI paths. Moreover, most std values of the bond lengths in the ICONGENI paths were distributed below the corresponding experimental values while many of those in the NGENI paths were higher than the experimental values. Subsequently, we investigated the backbone bond angle distributions (including $N{-}C_\alpha{-}C$, $C_\alpha{-}C{-}N$, and $C{-}N{-}C_\alpha$) of the simulated pathways in the same manner as described above for analysis of the bond length distributions. Similar to the results in the bond length distribution graphs, the avg values of the bond angles in the ICONGENI paths were densely distributed around the corresponding experimental values, and the std values were distributed close to zero compared to the NGENI pathways (**Fig 5**). These quantitative data commonly showed that the ICONGENI pathways were more likely to prevent unrealistic distortions in bond lengths and bond angles than were the NGENI pathways. On the other hand, there were exceptions to this principle, such as the small number of bond lengths in the

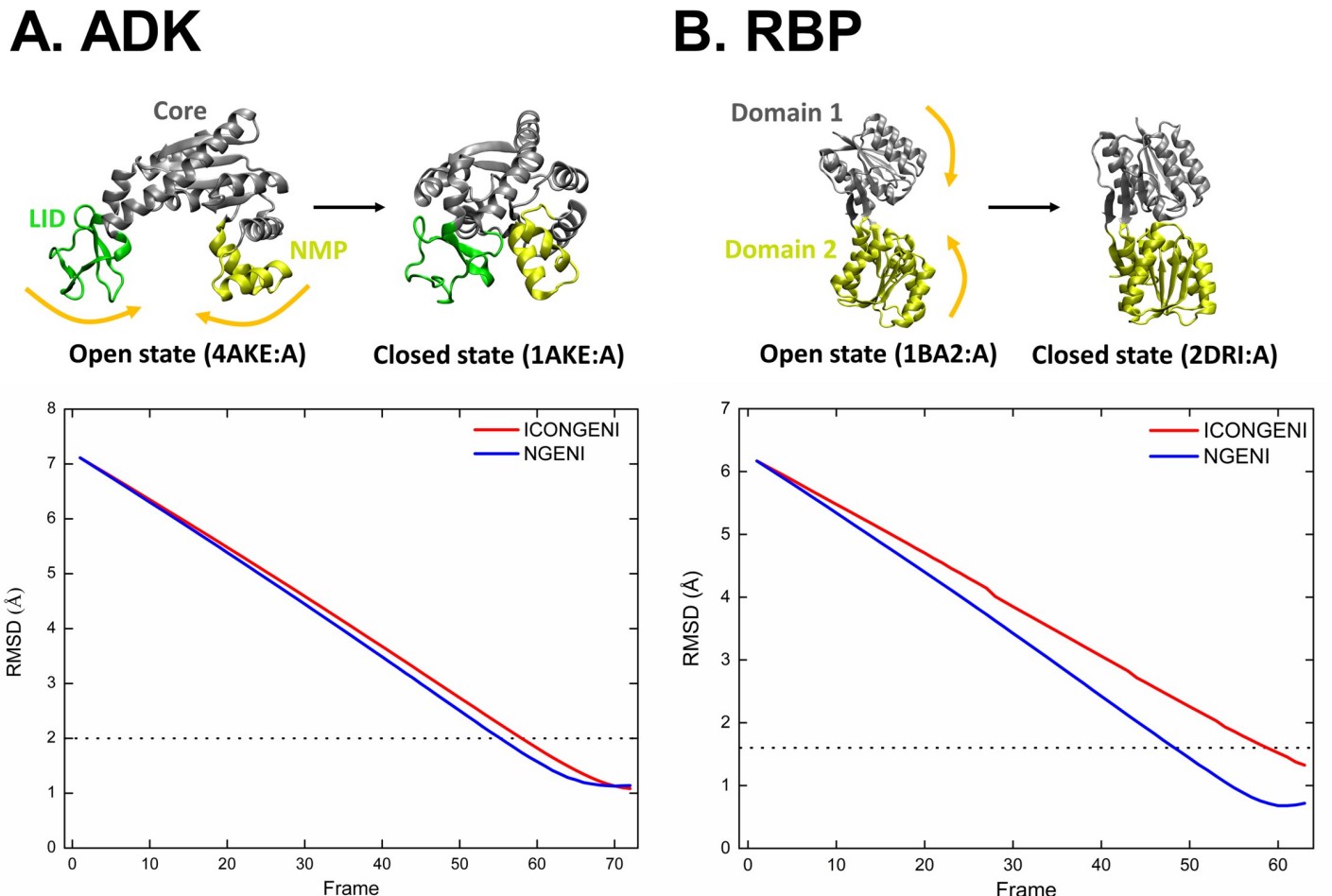

**Fig 3.** Conformational transition from open to closed states of (A) ADK and (B) RBP. Upper figures represent crystal structures of open and closed states of the proteins. ADK is composed of three domains: CORE (residues 1–29, 60–121, and 160–214), NMP (residues 30–59), and LID (residues 122–159). RBP is composed of Domain 1 (residues 1–103 and 236–264) and Domain 2 (residues 104–235 and 265–271). The lower graphs describe the convergence of simulated pathways of each protein by measuring changes in RMSD between predicted intermediates and the final structure. The results of ICONGENI and NGENI pathways are shown as red and blue lines, respectively. The black dotted lines represent the corresponding experimental resolution of the proteins.

ICONGENI paths with irrational avg or std values that deviated farther from the corresponding experimental ones than did those of the NGENI path (denoted by pink circles in **Fig 4B**). We confirmed that all bond lengths that fell under these exceptions belonged to either the first or last residue of the proteins, which would imply that these unrealistic distortions were caused by the tip effect [41]. The tip effect is an inherent weakness of NMA (regardless of the type of coordinate system) in which the highly flexible parts in protein structures (*e.g.*, hanging loops and protruding ends) exhibit abnormal behavior in some of the lowest normal modes. IC-NMA may suffer more from the tip effect than did CC-NMA because the mode shapes in IC-NMA is intrinsically limited in describing the movements of either the first or last residue where any torsion angle cannot be defined, which can explain the exceptions in **Fig 4B**. However, this limitation of ICONGENI is not a critical issue in predicting the transition pathways because the distortions of the tip parts could be considered as local vibrations that has little effect on the global motions.

In the same context as investigating the bond length and bond angle distributions, the potential energies of the intermediate conformations comprising the resulting pathways were

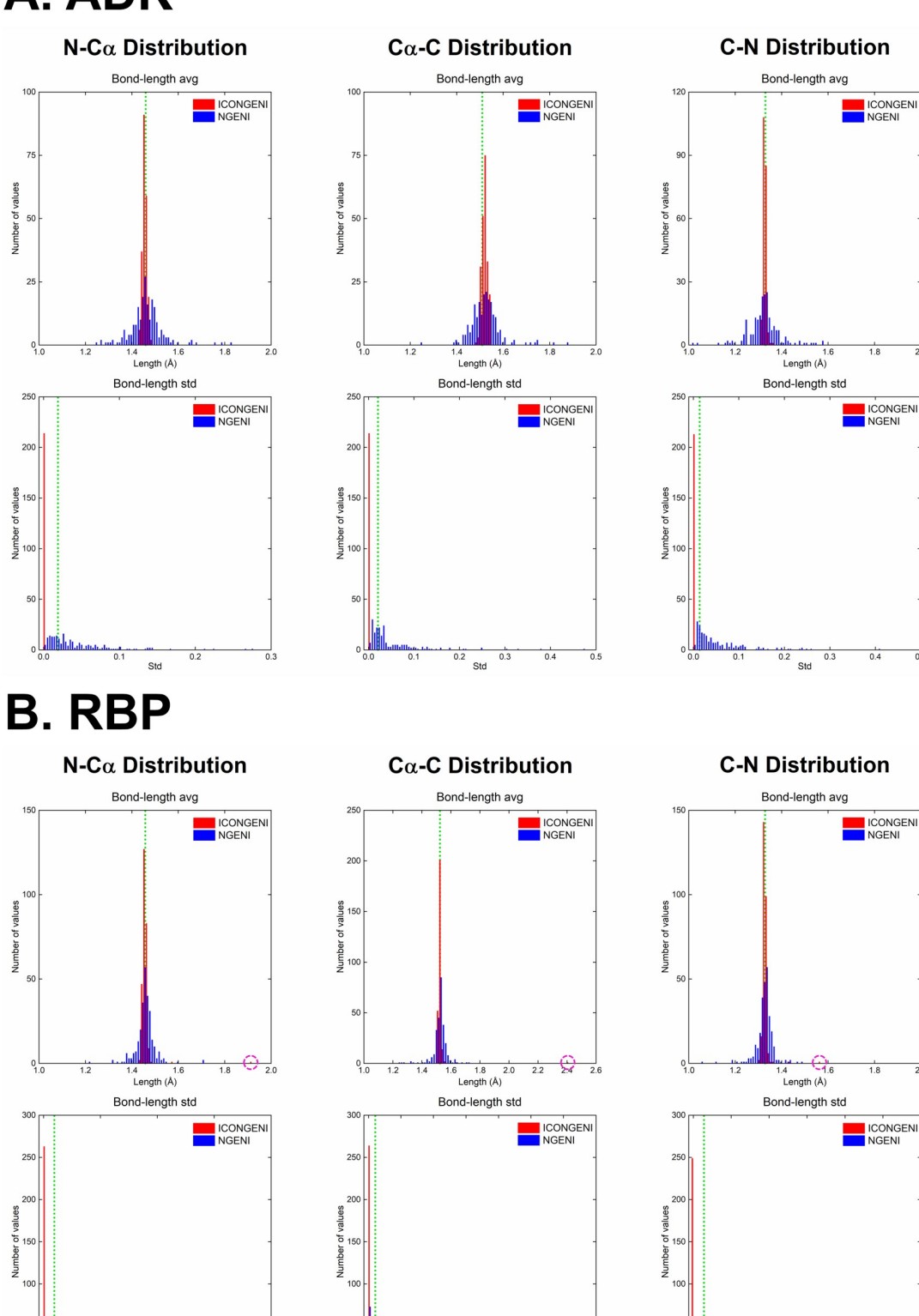

**Fig 4.** Bond length distribution of the transition pathways of (A) ADK and (B) RBP. Comparison of the distributions of the avg and std values of backbone bond lengths in ICONGENI (denoted by red) and NGENI (denoted by blue) pathways. Both methods explore the transition pathways based on the DOFs of the 50 lowest normal modes. The bond length distributions are measured for three backbone bond length coordinates: $N-C_\alpha$, $C_\alpha-C$, and $C-N$. The green lines represent the corresponding experimental values of the coordinates. The pink circles represent specific cases where the ICONGENI pathway has irrational values of bond length avg or std.

calculated. Because both simulation methods were carried out by using a coarse-grained modeling method, non-backbone atoms and O atoms in the backbone from reference structures (4AKE:A and 1BA2:A for ADK and RBP, respectively) were grafted to all intermediates and the generated all atom models were energy minimized within CHARMM36m force field for 500 steps of conjugate gradient [37] to eliminate any steric clashes and inappropriate geometries. Next, we calculated the potential energies of all intermediates of the NGENI and ICONGENI pathways by using CHARMM36m force field to quantitatively evaluate how the corresponding transitions are stable. **Fig 6** shows the difference of the potential energy of each frame between the NGENI and ICONGENI pathways. From the results, we confirmed that the ICONGENI pathways are generally more stable (*i.e.*, having lower potential energies) than the NGENI pathways. Furthermore, their energy gap increased as the pathways progress, which suggests that the qualitative difference between the pathways is increasingly noticeable in that the geometric errors are gradually accumulated when anharmonic transitions are explored by harmonic modes. Finally, these simulation results imply that the ICONGENI pathways are more reliable than the NGENI pathways in terms of thermal and chemical stability.

## 3.2 Predicting a transition pathway ensemble depending on a set of lowest normal modes

In the previous section, we tried to predict conformational transition pathways using ICONGENI with the 50 lowest normal modes considered sufficient to describe large deformation. Although the resulting pathways are shown to be reliable in terms of thermal and chemical stability, this study does not verify that ICONGENI can provide information on real transition trajectories. ICONGENI with the 50 lowest normal modes finds the deterministic and most effective pathways in terms of atomic displacements due to the intrinsic properties of the established cost function (see Section 2.3). In this section, we discuss the possibility that ICONGENI can predict plausible routes for conformational changes on complex energy landscapes by applying it to ADK of which transition mechanisms have been studied in numerous research works.

First, we generated an ensemble of the transition pathways of ADK through ICONGENI depending on the number of lowest normal modes (from 5 to 100), and their convergence was measured by RMSD with the closed state, as in the previous section. As shown in **Fig 7A**, the fewer are the normal modes used in the simulation, the less likely it is that the corresponding pathway reaches the final conformation. In detail, the pathways using fewer than 25 normal modes do not satisfy the convergence condition (*i.e.*, their RMSD from the final conformation does not converge under the experimental resolution of ADK). This is not surprising given that the progression of pathways is influenced strongly by the DOFs describing the structural motions. In addition, this result suggests that it is necessary to focus on the "incomplete" pathways simulated with relatively few normal modes, as molecular systems usually explore seemingly inefficient routes of conformational transitions to arrive at functional states over several high-energy barriers.

Although the detailed transition mechanisms of ADK remain to be elucidated, previous experimental and theoretical studies have proposed several pathways via the NMP-closing/

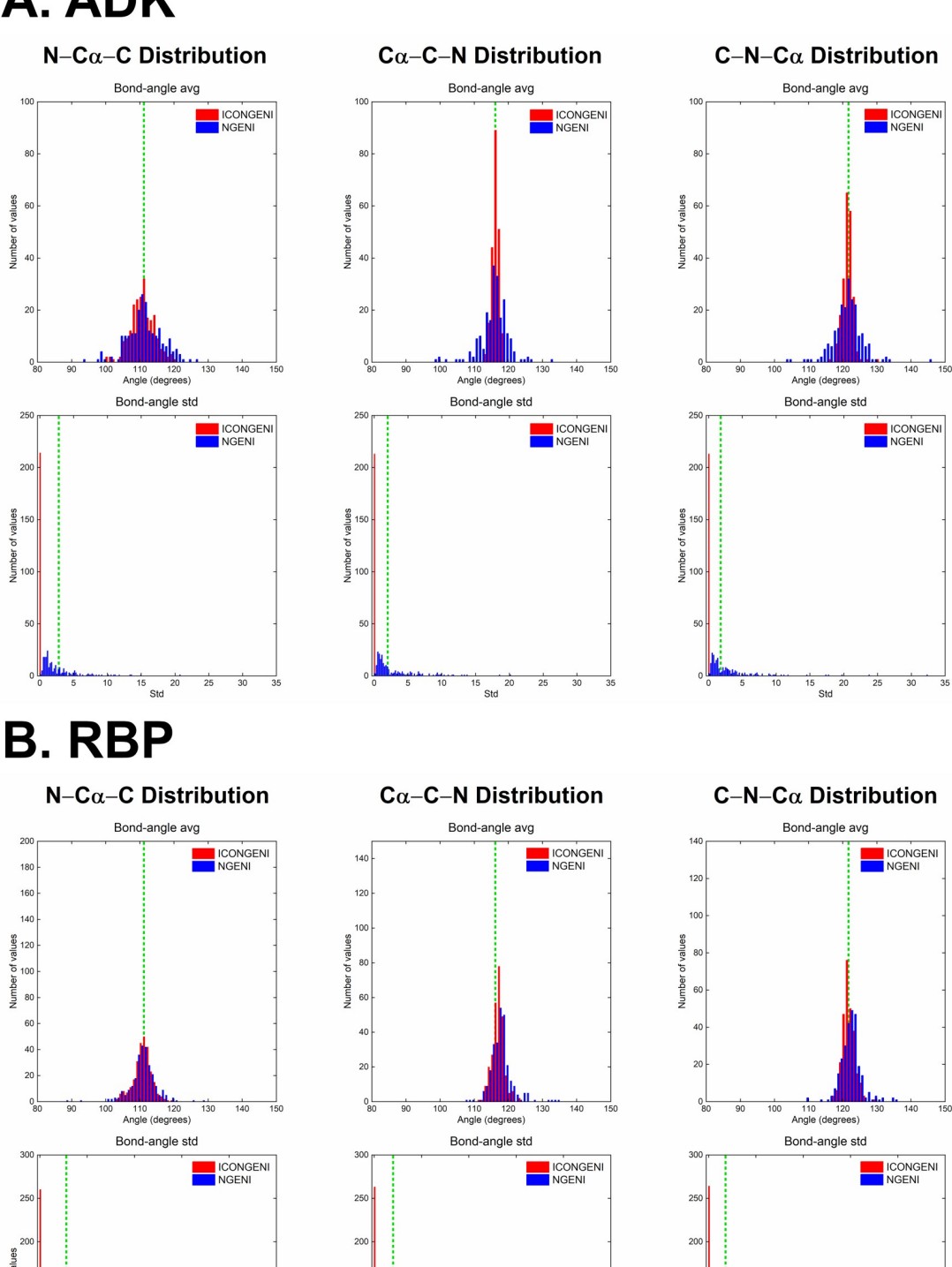

**Fig 5.** Bond angle distribution of the transition pathways of (A) ADK and (B) RBP. Comparison of the distributions of the avg and std values of backbone bond angles in ICONGENI (denoted by red) and NGENI (denoted by blue) pathways. Both methods explore the transition pathways based on the DOFs of the 50 lowest normal modes. The bond angle distributions are measured for backbone bond angle coordinates: $N–C_\alpha–C$, $C_\alpha–C–N$, and $C–N–C_\alpha$. The green lines represent the corresponding experimental values of the coordinates.

LID-opening ($NMP_C$) state or the NMP-opening/LID-closing ($NMP_O$) state [33, 34, 46–52]. In other words, the large-scale transition of ADK is characterized by interdomain hinge motions of NMP and LID relative to CORE. To delineate the NMP and LID movements on the ICONGENI pathways, we projected them onto interdomain angle space with the NMP-CORE angle ($\theta_{NMP}$) and the LID-CORE angle ($\theta_{LID}$). $\theta_{NMP}$ ($\theta_{LID}$) is defined by the centers of mass of the backbone including $N$, $C\alpha$, and $C$ in residues 115–125, 90–100, and 35–55 (179–185, 115–125, and 125–153) in the notation used in a previous study [51]. In addition, we used some experimental structures for cross-validation with our simulation results. 4AKE:A and 1AKE:A defines the two end-point conformations of the transition. The crystal structures whose PDB codes are 1ZIN, 1ZIO, and 1ZIP [33] and those whose PDB code is 1DVR [34] approximate the $NMP_C$ state and the $NMP_O$ state, respectively. On $\theta_{NMP}$–$\theta_{LID}$ space, the pathway ensemble has a tendency: the larger is the number of normal modes used in the simulation, the straighter are the resulting pathways to the closed state (**Fig 7B**). This is not surprising given that the established cost function is designed to produce the most efficient and direct paths within the given DOFs. The straight path on the interdomain angle space refers to the trajectory at which NMP and LID simultaneously open during the transition but is not favorable in terms of the free energy landscapes [47, 48]. As the number of modes used in the simulation decreased to less than 35, the resulting pathways tended to closely approach the $NMP_C$ state. When using significantly fewer normal modes (less than 10) for simulations, the corresponding pathways described the transition toward the more extreme $NMP_C$ state than did the

# A. ADK

# B. RBP

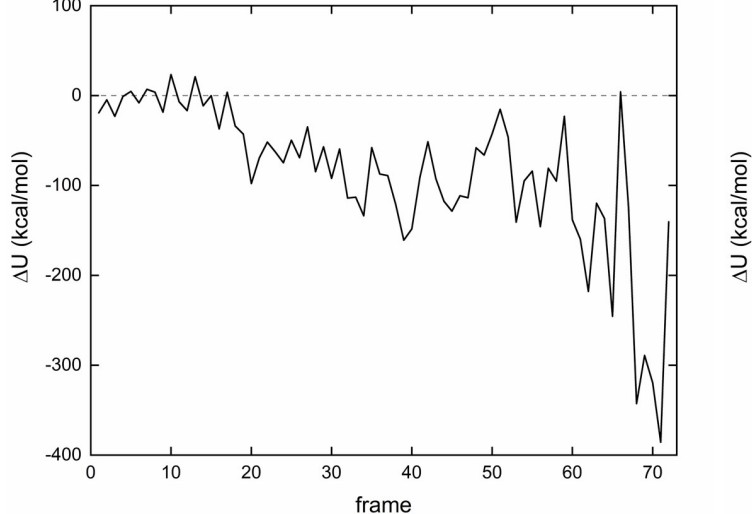
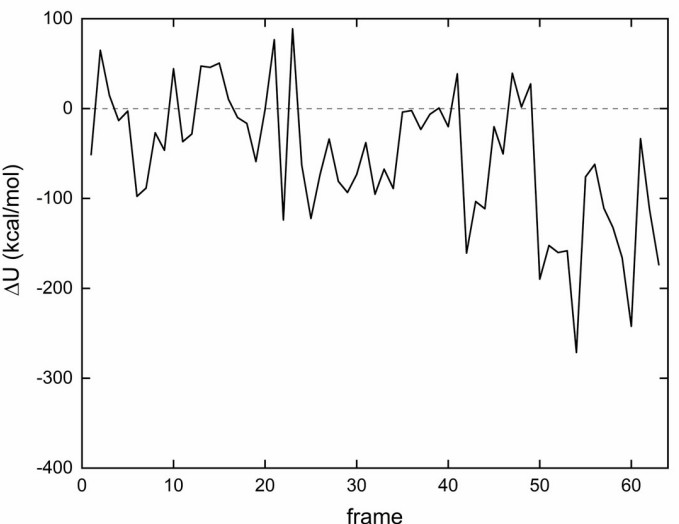

**Fig 6.** The difference between potential energies of ICONGENI and NGENI pathways for (A) ADK and (B) RBP. The difference of the potential energy $\Delta U = U_{ICONGENI} - U_{NGENI}$. Before calculating the potential energies, all simulated intermediate structures were transformed into all atom models based on corresponding reference structures (4AKE:A (1BA2:A) for ADK (RBP)) and were energy minimized using CHARMM36m force field for 500 steps of conjugate gradient.

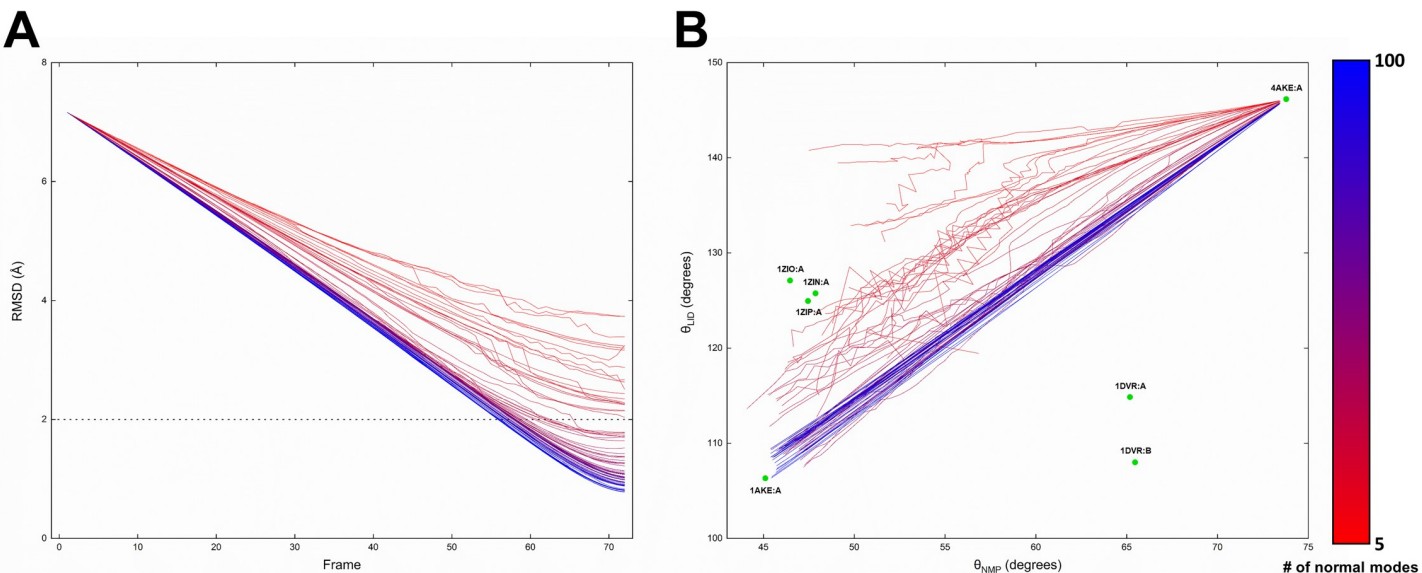

**Fig 7. The pathway ensemble for ADK generated by ICONGENI.** The ICONGENI transition pathways that make up the pathway ensemble are colored according to the lowest normal modes (from 5 to 100) used in the simulation (red to blue color scheme). (A) Convergence of the pathway ensemble. The RMSD values of each path relative to a final state are measured. The black dotted line represents the corresponding experimental resolution of ADK. (B) Projection of the pathway ensemble onto $\theta_{NMP}$–$\theta_{LID}$ space. The green points show the positions of experimental structures on $\theta_{NMP}$–$\theta_{LID}$ space. 4AKE:A and 1AKE:A indicate the open and closed states of ADK, respectively. 1ZIN:A, 1ZIO:A, and 1ZIP:A (1DVR:A, and 1DVR:B) indicate experimental structures at the $NMP_C$ state (the $NMP_O$ state).

crystal structures approximating the $NMP_C$ state (*i.e.*, 1ZIN:A, 1ZIO:A, and 1ZIP:A). This result implies that the vibrational features describing the dynamics of $\theta_{NMP}$ were preferentially arranged in the lowest normal modes, resulting from the flexibility between NMP and CORE is higher than that between LID and CORE. Therefore, we suggest that the open-to-closed transition via the $NMP_C$ state is more plausible and reliable than that via the $NMP_O$ state in terms of the vibrational characteristics of ADK. However, this result does not mean ICON-GENI always returns a single candidate (*i.e.*, paths via the $NMP_C$ state) of the transition paths. If the normal mode set as the system DOFs is determined under certain conditions, ICON-GENI could explore transition pathways via the $NMP_O$ state, which demonstrate that ICON-GENI can explore multiple transition pathways compatible to several metastable states if information of the states is given (See more details in **S1 Text** and **S1 Fig**).

## 4 Conclusion

In this study, we introduced internal coordinate normal mode-guided elastic network interpolation (ICONGENI) as a theoretical method to explore the conformational transition pathways of proteins. By linearly interpolating the coarse-grained models of the two end-point states, ICONGENI defines virtual intermediate conformations of which the transition pathway is composed. Based on structural information, ICONGENI explores the optimal transition pathway (*i.e.*, the pathway minimizing a cost function showing the error between the simulated intermediates and the virtual ones). When iteratively obtaining the consecutive conformations describing the transition pathway, the key idea of the method is to represent the displacement vectors as a linear combination of lowest normal mode vectors produced by normal mode analysis in internal coordinates (IC-NMA). Given that IC-NMA can describe chemically relevant dynamics (suitable for describing large-scale transitions) compared to NMA in Cartesian coordinates (CC-NMA), this strategy enables the proposed method to explore reliable transition pathways in an efficient manner.

To evaluate the superiority of ICONGENI, we performed comparative studies of ICON-GENI with our previous method based on CC-NMA (named NGENI). For two proteins: adenylate kinase (ADK) and ribose-binding protein (RBP), we predicted transition pathways through the two methods under the same conditions (using the 50 lowest normal modes as the system degrees of freedom). The distribution data of the bond lengths and bond angles of the resulting pathways confirmed that these coordinates remained highly stable in the ICONGENI pathways compared to those in the NGENI pathways (**Figs 4** and **5**). Furthermore, we also calculated the potential energies of the simulated pathways and identified the energies of the ICONGENI pathways were lower overall than those of the NGENI pathways (**Fig 6**). In conclusion, these results suggest that IC-NMA is suitable for representing realistic dynamics of the proteins, by extension, that ICONGENI could explore more reliable transition pathways than NGENI in terms of thermal and chemical stability.

Although ICONGENI using the degrees of freedom (DOFs) of the 50 or more lowest normal modes can provide a spatial understanding of conformational transitions, this approach is insufficient to explain the actual transition events on complex energy landscapes. To address this issue, we focused on a pathway ensemble for ADK simulated by ICONGENI. First, we confirmed that the more is the number of normal modes used in the simulation, the closer the initial structure is to the final one, which is not surprising because the number of normal modes directly indicates the DOFs to describe structural dynamics (**Fig 7A**). Next, we characterized the pathway ensemble by interdomain angles of ADK (*i.e.*, the NMP-CORE angle ($\theta_{NMP}$) and the LID-CORE angle ($\theta_{LID}$)) and found that the deficient pathways (using less than 50 lowest normal modes) provided meaningful insights into the conformational transitions of ADK. When projecting the ensemble onto $\theta_{NMP}-\theta_{LID}$ space, the deficient pathways showed the conformational transitions toward a metastable intermediate state (*i.e.*, the NMP-closing/LID-opening state) while the sufficient pathways (using more than 50 lowest normal modes) showed those directly to the final state (*i.e.*, the closed state) with unrealistic deformation (**Fig 7B**). Therefore, it is concluded that ICONGENI can explore meaningful transition pathways on complex energy landscapes.

The key role of computational approaches in investigating conformational transitions of proteins is to predict the trajectories that are beyond experimental capabilities. Our technique outlined here can shed light on the transition mechanisms in an efficient manner using only information on experimentally observed end-point structures. Furthermore, the simulation results strongly depend on a set of low-frequency normal modes as the system DOFs, enabling the method to generate a pathway ensemble based on dynamic characteristics and to provide low-energy paths. In this regard, our technique has the potential to find good candidates of unknown intermediate states on complex energy landscapes.

## Supporting information

**S1 Text. The ICONGENI simulations to explore ADK transition mechanisms via the NMP$_O$ state.**
(DOCX)

**S1 Fig. The transition pathways for ADK via the NMP$_O$ state generated by ICONGENI.** Seven pathways in the vicinity of the NMP$_O$ state are shown on $\theta_{NMP}-\theta_{LID}$ space. The pathway named "diff. *a* degrees" means that it was explored by ICONGENI using the lowest normal modes that satisfy the condition: $\Delta\theta_{LID}-\Delta\theta_{NMP}>a$ (see details in **S1 Text**). The ADK crystal structures are taken as the references (indicated by green circles). 4AKE:A and 1AKE:A indicate the open and closed states of ADK, respectively. 1ZIN:A, 1ZIO:A, and 1ZIP:A (1DVR:A, and 1DVR:B) indicate experimental structures at the NMP$_C$ state (the NMP$_O$ state). The

pathway ensemble data (Fig 7B) is also included in this figure for comparison.
(TIF)

**S1 Movie. The transition pathway for ADK simulated by ICONGENI based on the DOFs of the 50 lowest normal modes.**
(WMV)

**S2 Movie. The transition pathway for RBP simulated by ICONGENI based on the DOFs of the 50 lowest normal modes.**
(WMV)

## Author Contributions

**Conceptualization:** Byung Ho Lee, Soon Woo Park, Soojin Jo, Moon Ki Kim.

**Data curation:** Byung Ho Lee, Moon Ki Kim.

**Formal analysis:** Byung Ho Lee, Soon Woo Park.

**Funding acquisition:** Moon Ki Kim.

**Investigation:** Byung Ho Lee.

**Methodology:** Byung Ho Lee, Soon Woo Park, Moon Ki Kim.

**Project administration:** Moon Ki Kim.

**Resources:** Moon Ki Kim.

**Software:** Byung Ho Lee, Soon Woo Park.

**Supervision:** Moon Ki Kim.

**Validation:** Byung Ho Lee, Moon Ki Kim.

**Visualization:** Byung Ho Lee.

**Writing – original draft:** Byung Ho Lee.

**Writing – review & editing:** Byung Ho Lee, Soon Woo Park, Soojin Jo, Moon Ki Kim.

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
