## [Decision Letter · Decision Letter 0]

7 Jul 2021

PONE-D-21-19500

Protein Conformational Transitions Explored by a Morphing Approach Based on Normal Mode Analysis in Internal Coordinates

PLOS ONE

Dear Dr. Kim,

Thank you for submitting your manuscript to PLOS ONE. After careful consideration, we feel that it has merit but does not fully meet PLOS ONE’s publication criteria as it currently stands. Therefore, we invite you to submit a revised version of the manuscript that addresses the points raised during the review process.

We look forward to receiving your revised manuscript.

Kind regards,

Wenfei Li, Ph.D.

Academic Editor

PLOS ONE

Journal Requirements:

Reviewers' comments:

Reviewer's Responses to Questions

**Comments to the Author**

1. Is the manuscript technically sound, and do the data support the conclusions?

Reviewer #1: Partly

Reviewer #2: Partly

Reviewer #3: Yes

2. Has the statistical analysis been performed appropriately and rigorously? 

Reviewer #1: Yes

Reviewer #2: I Don't Know

Reviewer #3: N/A

3. Have the authors made all data underlying the findings in their manuscript fully available?

Reviewer #1: Yes

Reviewer #2: Yes

Reviewer #3: Yes

4. Is the manuscript presented in an intelligible fashion and written in standard English?

Reviewer #1: Yes

Reviewer #2: Yes

Reviewer #3: Yes

5. Review Comments to the Author

Reviewer #1: The authors proposed an interesting path-sampling method to when end-point structures are given a priori. This method was developed based on a morphing algorithm and the normal mode analysis (NMA) and allows us to generate intermediate states between the end-point structures with low computational costs. The theoretical part was organized well by referring to their previous studies for this extension study. However, there might be some issues to be addressed prior to actual applications. Therefore, I would like to ask the following issues before publishing.

1) This method provides a set of (or an ensemble) of transition paths between given end-point structures. However, it is difficult to directly calculate free energy profiles based on this method when evaluating generated paths. In actual applications, users want to evaluate generated transition paths quantitatively on their free energy profiles. Herein, how to evaluate the generated paths quantitively using some physical quantities? Are there additional treatments for the quantitative evaluations? I would like to know a way to validate the generated transition paths.

2) In this study, the authors specified the 50 lowest modes as DOFs for the linier combinations. Why did you specify these modes? Are there any conditions to specify optimal DOFs to generate reliable transition paths? I think that users would like to know a way to specify the optimal DOFs.

3) Does this method work if there are multiple transition paths between given end-point structures? Does the current strategy return one of the multiple transition paths? Does this method always return a single candidate of transition paths? Or can this method always return the minimum free energy path (MFEP)?

4) For Eq ([Disp-formula pone.0258818.e038]), a target intermediate is determined by the linear interpolation between given end-point structures. Is this assumption enough to describe complicated transition paths of proteins? My concern is that this simple linear interpolation might include some artifact when we must consider complicated transitions of proteins between given end-point structures.

5) Related to previous issue (the 3rd issue), did this method successfully detect multiple transition paths of AK? In previous studies, several enhanced sampling methods detected multiple transitions path between the open and closed states of AK (for example, J. Chem. Theory Comput, 2019, 15, 5199-5208, Fig. 5). I would like to a correspondence of your result with previous study’s results.

6) Why did you adopt the ENM potential as a coarse-grained (CG) potential? Are there any possibilities to specify the other CG potentials? Generally, I think that the ENM potential is used to describe harmonic modes around a relaxed configurations using a reference structure such as an experimental (X-ray crystal or NMR) structure. Therefore, I think that this ENM potential might be unsuitable for describing large-amplitude domain motions. Why can this ENM potential treat or induce anharmonic motions of AK or RBP?

7) I think there might be a typo on Eq ([Disp-formula pone.0258818.e027]). It seems that the chain rule on Eq ([Disp-formula pone.0258818.e027]) does not work correctly in the present description.

Reviewer #2: Comments for the Authors

Lumping protein structures are undergoing large-scale conformational changes, which are essential to associate with their function at the cell and organism level, but have been elusive both experimentally and computationally. In this study, Lee et al. proposed a new computational methodology called internal coordinate normal mode-guided elastic network interpolation (ICONGENI) to sample intermediate conformations by interpolating the interatomic distance between two end-point conformations with the degrees of freedom constrained by the low-frequency dynamics afforded by normal mode analysis (NMA) in internal coordinates. The applications to two case studies (adenylate kinase (ADK) and ribose-binding protein (RBP)) by analyzing their open-to-closed transitions show that ICONGENI can explore highly reliable pathways in terms of chemical and thermal stability. Take together, the method illustrated in this study exhibits a certain degree of potentiality in terms of intermediated conformations predictions for these proteins with large-scale conformational changes during functioning. This study is a continuous work to the previous one, basically by changing the position descriptor of cartesian coordinates to that of internal coordinates. However, it’s superiority was not emphasized enough by the way of presenting/writing in this manuscript.

There are several points described below,

1. What are the computational costs for ICONGENI and NGENI methods? Is ICONGENI method much more computationally cheaper than NGENI? I would suggest to list limits and merits of two methods in a table.

2. The energy minimization details should be provided. Such as what is the minimization method, how many steps.

3. What is the cutoff distance used in the elastic network model (ENM)? What is the correlation between this parameter and the predicted results?

4. “For both proteins, the avg (std) values in the ICONGENI pathways were more concentrated around (below) the experimental values than were those in the NGENI pathways (Fig. 4).” Here the authors should provide more elaborate discussion regarding the reason behind this distribution difference.

5. “While the simulation result for RBP showed that ICONGENI suffered more from the tip effect than did NGENI, this was not a critical issue because the distortions of the tip parts could be considered local vibrations that barely contributed to the conformational transitions” Why did ICONGENI suffer more from the tip effect than did NGENI? Is this a limitation for this method potentially?

6.Are the structures of the intermediated states along the predicted pathways extractable? If so, their structures should be presented (if too many intermediated structures, the authors can make a movie). If not, why?

7. In Fig.3, “The black dotted lines represent the corresponding experimental resolution of the proteins.” Sorry, I don’t understand the description here. As the vertical axis is RMSD. I also didn’t quite understand the meaning of frame in the horizontal axis. Are they predicted intermediated conformations? Why did authors use ~ 70 frames for ADK but ~ 60 frames for RBP?

8. Page 22, line 425, a “than” should be added after “reliable”.

9.The color bar used in the Fig.6 is very difficult to differentiate, the rainbow color scheme might be better.

10. The superiority was not emphasized enough by the way of presenting/writing in this manuscript. More analyses and comparisons should be conducted.

11.Is the link of code available? If so, I suggest to provide the link in Abstract.

Reviewer #3: This manuscript describes a new methodology based on Normal Mode Analysis framework to decipher protein conformational transitions by quantitatively capturing often elusive intermediate conformational states. While the methodology has merit, I have several major comments that the authors must address to render the manuscript suitable for publication in PLOS ONE.

1. The authors describe in the Introduction (line 51-53) section that "MD simulation has some problems (like those of experimental methods)". This sentence is written rather vaguely because the context of comparing MD methodology to numerous experimental techniques has to be clearly laid out, which has not been done by the authors. The authors must provide context and provide clear contextual justification when they are comparing the technical details (which are distinct) in experimental techniques and computational techniques, including MD simulation.

2. The authors must elucidate (line 73-74) regarding the novelty of NMA obtained in Internal Coordinates (IC). How does the choice/assignment of IC affect the underlying analysis framework? The authors should comment on this.

3. Line 111: the authors must clearly state what they mean by conformational energy and how it is assigned to a conformational state.

4. Line 116: it is needed to justify why the authors chose DOPE score to select the best models corresponding to each template? Were any other scoring functions used? If not, then the authors must clearly weigh in on the merit of using the DOPE score as a selection criteria of their models.

5. Lines 127-128: The authors should comment on why curvilinear coordinates do not have a more significant impact on the relative spatial orientation of intermediate states.

6. The sentence in lines 153-154 should be re-written for clarity.

7. The authors should comment on why considering bond lengths and bond angles to be fixed does not affect the framework. While it is an approximation in their model, more quantitative justification is needed.

8. In Figure 1, the authors should comment on the validity of considering the two domains, A and B, as rigid bodies, and not consider local/distal fluctuations.

9. How does comparing the two approaches (previously developed, NGENI) and the current model (ICONGENI) ensure that the pathways considered are the same? Even setting all physical quantities in the respective calculations to be the same, how is it inherently ensured that they sample the same pathway? The authors must comment on/clarify this.

10. Lines 410-411 : The authors should clearly state what experimental structures were chosen for their experimental check? This sentence is ambiguous.

11. The authors should comment on how do they ensure that given two end point conformations, they are sampling the same pathway in multiple iterations. How will the framework apply if they are considering open/closed transition of a protein that has metastable intermediate states which may be sampled at certain physical conditions?

12. The Conclusion section seems to have a broad overlap with the Materials and Methods section, and as such it needs to be re-organized The authors must attempt to provide a comprehensive evaluation of the merits and limitations of their method when summarizing their models and its applicability to different types of systems.

In addition, the figures must be re-formatted as they are of rather poor resolution.

6. PLOS authors have the option to publish the peer review history of their article (what does this mean?). If published, this will include your full peer review and any attached files.

Reviewer #1: No

Reviewer #2: No

Reviewer #3: **Yes: **Prithviraj Nandigrami

---

## [Author Response · Author response to Decision Letter 0]

28 Aug 2021

First of all, we thank the referees for providing thoughtful and helpful suggestions and comments. Our responses to the comments are included in an uploaded file named "Response to Reviewers".

---

## [Decision Letter · Decision Letter 1]

6 Oct 2021

Protein Conformational Transitions Explored by a Morphing Approach Based on Normal Mode Analysis in Internal Coordinates

PONE-D-21-19500R1

Dear Dr. Kim,

We’re pleased to inform you that your manuscript has been judged scientifically suitable for publication and will be formally accepted for publication once it meets all outstanding technical requirements.

Kind regards,

Wenfei Li, Ph.D.

Academic Editor

PLOS ONE

Additional Editor Comments (optional):

This work reports a computational method for sampling the transition pathways of proteins between two functional structures, which can be useful in the studies of protein functional dynamics. Based on my reading and the recommendations by three referees, I am happy to recommend its acceptance for publication.

Reviewers' comments:

Reviewer's Responses to Questions

**Comments to the Author**

1. If the authors have adequately addressed your comments raised in a previous round of review and you feel that this manuscript is now acceptable for publication, you may indicate that here to bypass the “Comments to the Author” section, enter your conflict of interest statement in the “Confidential to Editor” section, and submit your "Accept" recommendation.

Reviewer #1: All comments have been addressed

Reviewer #2: All comments have been addressed

Reviewer #3: All comments have been addressed

2. Is the manuscript technically sound, and do the data support the conclusions?

Reviewer #1: Partly

Reviewer #2: Yes

Reviewer #3: Yes

3. Has the statistical analysis been performed appropriately and rigorously? 

Reviewer #1: Yes

Reviewer #2: Yes

Reviewer #3: N/A

4. Have the authors made all data underlying the findings in their manuscript fully available?

Reviewer #1: Yes

Reviewer #2: Yes

Reviewer #3: Yes

5. Is the manuscript presented in an intelligible fashion and written in standard English?

Reviewer #1: Yes

Reviewer #2: Yes

Reviewer #3: Yes

6. Review Comments to the Author

Reviewer #1: The authors have carefully addressed all the issues pointed by the reviewers. The revised manuscript is publishable.

Reviewer #2: All of my previous comments have been addressed throughly. Thus, I would like to recommend the direct publication of the current version.

Reviewer #3: The authors have adequately addressed my concerns and comments. I would like to recommend publication of the manuscript.

One minor (optional) point I would like to mention: the authors should attempt to provide a comprehensive account of the merits and limitations of their current method, and compare it against similar method(s) available in the literature. This would most certainly help the readers weigh the applicability of the account presented in this manuscript more thoroughly.

7. PLOS authors have the option to publish the peer review history of their article (what does this mean?). If published, this will include your full peer review and any attached files.

Reviewer #1: No

Reviewer #2: **Yes: **Wei Liu

Reviewer #3: No

---

## [Editor Report · Acceptance letter]

27 Oct 2021

PONE-D-21-19500R1 

Protein Conformational Transitions Explored by a Morphing Approach Based on Normal Mode Analysis in Internal Coordinates 

Dear Dr. Kim:

I'm pleased to inform you that your manuscript has been deemed suitable for publication in PLOS ONE. Congratulations! Your manuscript is now with our production department. 

Kind regards, 

on behalf of

Dr. Wenfei Li 

Academic Editor

PLOS ONE